

# Low and Consistent Asymmetry Parameters in Arctic and Mid-latitude Cirrus

Emma Järvinen[1] and Franz Martin Schnaiter[1,2]

[1]Institute for Atmospheric and Environmental Research, University of Wuppertal, Wuppertal, Germany
[2]schnaiTEC GmbH, Wuppertal, Germany

**Correspondence:** Emma Järvinen (jaervinen@uni-wuppertal.de)

**Abstract.** Cirrus clouds play a critical role in the Earth's radiation budget, yet their shortwave optical properties remain poorly constrained. The asymmetry parameter ($g$), which governs the angular distribution of scattered light, is particularly sensitive to ice crystal morphology, a property that varies widely in cirrus. To provide observational constraints on the magnitude of $g$ and to investigate its relationship with ice microphysical properties, we analysed simultaneous in situ measurements of

particle morphology and angular light scattering using the Particle Habit Imaging and Polar Scattering (PHIPS) probe. These measurements were conducted during the Cirrus in High Latitudes (CIRRUS-HL) campaign in June and July 2021, which sampled both mid-latitude and Arctic cirrus across a range of cloud types and temperatures down to –63°C. We found that cirrus in both regions exhibited a consistently low median asymmetry parameter of 0.727. The observed $g$ values were largely insensitive to variations in temperature, humidity, and crystal aspect ratio, and showed only minor variation across ice habits. A

systematic decrease in $g$ with increasing particle size was identified, ranging from 0.760 for sub-30 µm particles in mid-latitude cirrus to minimum values of 0.707 and 0.703 for 175 µm particles in mid-latitude and Arctic cirrus, respectively. The measured values are significantly lower than those commonly used in current radiative transfer schemes, suggesting that cirrus clouds may contribute less to net atmospheric warming than often assumed. These results provide improved observational constraints for the representation of ice cloud optical properties in climate models and support efforts to reduce uncertainties in cirrus

cloud radiative forcing.

## 1    Introduction

To improve the representation of bulk radiative properties of cirrus clouds in climate models and remote sensing algorithms, it is important to have a good understanding of the ice crystal single-scattering properties (Mishchenko et al., 1996; Zhang et al., 1999; Yang et al., 2001). The key single-scattering properties used in a two-stream radiative transfer formalism are

the extinction coefficient, single-scattering albedo and the asymmetry parameter ($g$), from which the latter has a complicated relationship with ice crystal morphology in the solar spectrum. Ice crystal morphological properties, like crystal size, habit as well as degree and type of crystal complexity, have been observed to vary greatly between different cloud types, in-cloud temperature, and geographical location (e.g. Heymsfield et al., 2017; Wolf et al., 2018; Woods et al., 2018; Lawson et al., 2019).



Due to this high morphological variability, constraining the single-scattering properties of ice crystals in numerical simula-
tions remains a major challenge. Numerical ray-tracing studies for the solar spectral range predict a wide variety of $g$ values,
typically between 0.74 and 0.94, with higher values associated with pristine hexagonal habits and lower values with complex
particle shapes or surface roughness (e.g., Iaquinta et al., 1995; Macke et al., 1996b, 1998; Um and McFarquhar, 2007; Yang
et al., 2008). However, it is important to correctly quantify $g$ as biases can lead to a significant uncertainty in estimations of the

ice cloud radiative effect in climate models (Järvinen et al., 2018; Yi et al., 2013) or in retrieved cirrus microphysical properties
from remote sensing observations (Yang et al., 2008; Yi et al., 2017a). Vogelmann and Ackerman (1995) estimated that $g$ must
be known within 2-5% in order to constrain computed shortwave fluxes to within approximately 5%.

Observational studies are needed to constrain the magnitude of $g$ in radiative transfer simulations. Unlike numerical esti-
mates of the ice crystal asymmetry parameter, measurements indicate a more moderate variability in the cirrus optical prop-

erties. Studies estimating the asymmetry parameter from radiometric observations have reported a low asymmetry parameter
around 0.7 in a case study over the midwest USA (Stephens et al., 1990) and previous satellite-based multi-angle observations
have reported values for the visible asymmetry parameter below 0.8 (van Diedenhoven et al., 2013; Wang et al., 2014; van
Diedenhoven et al., 2014; Cole et al., 2014; van Diedenhoven, 2021). Similarly, lidar observations provide another means of
retrieving the asymmetry parameter, with recent studies also reporting values around 0.76 (Gil-Díaz et al., 2025).

The limitation of estimating $g$ from remote sensing observations is their dependence on an optical model. The retrieved
values are inherently constrained by the model, meaning they can never be lower than the smallest $g$ allowed by the model. As
a result, retrievals frequently converge on this lower limit (Järvinen et al., 2023). For example, studies that identify the severely
roughened hexagonal aggregate model as the best fit for observations consistently retrieve $g$ values around 0.75 in the visible
(Cole et al., 2014; Järvinen et al., 2018; Wang et al., 2018; Forster and Mayer, 2022)

A more direct approach to measure $g$ is by using a polar nephelometer. So far, measurements of three airborne polar neph-
elometers have been reported in the literature: the Polar Nephelometer (PN; Gayet et al., 1998), the Cloud Integrating Neph-
elometer (CIN; Gerber et al., 2000), and the Particle Habit Imaging and Polar Scattering (PHIPS) probe (Abdelmonem et al.,
2016; Schnaiter et al., 2018). These measurements have shown median or average $g$ values between 0.73 and 0.79 (Gerber
et al., 2000; Auriol et al., 2001; Garrett et al., 2001, 2003; Gayet et al., 2004, 2006; Febvre et al., 2009; Gayet et al., 2012;

van Diedenhoven et al., 2013). However, direct measurements of $g$ have been limited to only a few airborne campaigns and
to mostly tropical deep convective or midlatitude systems. So far, the only in situ measurement reported in Arctic cirrus has
been one case study of a cirrus spissatus band with a cloud top temperature of -46°C (Garrett et al., 2001). Moreover, prior
studies have typically focused on bulk optical properties, with limited information on how $g$ relates to simultaneously measured
microphysical properties of the same ice crystal population.

The objective of this study is to expand the direct in situ observations of $g$ in cirrus by analyzing a comprehensive, multi-flight
dataset from both mid-latitude and Arctic cirrus, collected during the CIRRUS in High Latitudes (CIRRUS-HL) campaign
using the PHIPS instrument. This dataset enables a unique dual-regional comparison within the same season and includes
single-particle measurements that allow for a direct linkage between ice crystal microphysical properties and their angular light-
scattering characteristics. Beyond quantifying the magnitude of $g$, this study investigates its dependence on key microphysical





parameters, including effective radius, ice water content, habit, and aspect ratio. By combining bulk and particle-resolved analyses, the results provide new observational constraints for developing physically consistent optical parameterisations for radiative transfer applications.

## 2 Methodology

### 2.1 The CIRRUS-HL airborne campaign

The CIRRUS-HL airborne campaign took place from 6 June to 28 July 2021 using the High Altitude and Long Range (HALO) research aircraft (Jurkat-Witschas et al., 2025). The original aim of the campaign was to investigate the microphysical and optical properties of Arctic cirrus using in situ and remote sensing instrumentation, with operations initially planned from Kiruna, Sweden. However, due to pandemic-related restrictions, the campaign base was relocated to Oberpfaffenhofen, Germany. This logistical change broadened the scientific scope to include both mid-latitude and Arctic cirrus (see Fig. 1), as Arctic flight

operations now required long transit legs and refueling stops in Scandinavia or Iceland. The revised campaign strategy focused on contrasting cirrus clouds in different regions and meteorological conditions. Accordingly, flights were designed to sample natural mid-latitude cirrus in diverse weather regimes, including warm conveyor belt systems, high-pressure in situ cirrus, and convective outflow, as well as cirrus potentially influenced by aviation emissions. The time frame of the campaign coincided with pandemic air traffic restrictions, resulting in a 65 to 80% reduction in daily commercial flight activity compared to the

same time frame in 2019 (De La Torre Castro et al., 2023). The campaign period also overlapped with extreme rainfall events in central Europe, resulting in fatal flooding in Germany and Belgium (Lehmkuhl et al., 2022).

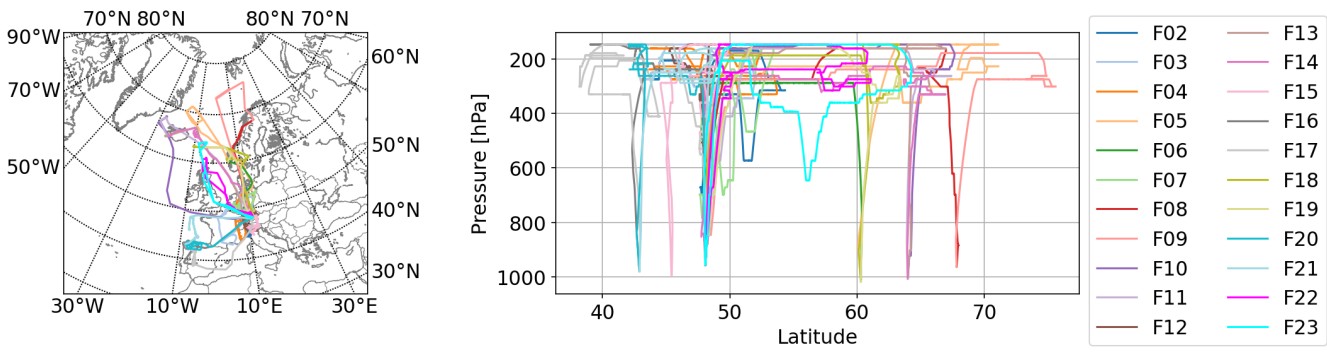

**Figure 1.** Trajectories of CIRRUS-HL measurement flights (F02 - F23) included to the analysis. Research flights targeting Arctic cirrus were performed as double flights with a refuelling stop. These were F05/F06, F08/F09 and F10/F11.

### 2.2 Microphysical and optical measurements

During CIRRUS-HL in situ measurements of the ice crystal microphysical and optical properties were conducted with the PHIPS airborne cloud instrument (Abdelmonem et al., 2016; Schnaiter et al., 2018). PHIPS is a combination of a single-





particle stereo-microscopic imager and a polar nephelometer, which allow for a combined analysis of optical and microphysical properties at the level of individual ice crystals.

### 2.2.1 Derivation of microphysical properties from stereo-microscopic images

The PHIPS stereo-microscopic imager consists of two CCD camera and microscope assemblies with an angular viewing distance of 120° each acquiring 1360 by 1024 pixel bright field microscopic images of individual cloud particles at a maximum

acquisition rate of 10 Hz. During CIRRUS-HL a magnification setting of 6x was used for both camera assemblies corresponding to an optical resolution of about 6 μm and a field of view of approximately $1.1{\times}1.5$ mm, limiting the maximum observable particle size to about 1.5 mm. For each image pair a set of microphysical parameters were retrieved using an algorithm discussed in Schön et al. (2011). These microphysical parameters include the maximum dimension ($D_{\max}$), area-equivalent diameter (i.e., equivalent diameter of a sphere having the same cross-sectional area, $D_{eq}$), projected area ($A_{proj}$),

and the aspect ratio ($AR$).

The stereo-microscopic images were also manually inspected and assigned to a habit class. A detailed description of the habit classification scheme can be found in the supplement of Järvinen et al. (2023). While the original habit classification scheme includes 18 categories, for the purposes of this study we reduced the habit classes to 9 cirrus-relevant classes. These include: *plate* (encompassing also sectored and skeleton plates), *column*, *frozen droplet*, *side plane*, *bullet rosette*, *mixed rosette*,

*capped column*, and *other*, which contains crystals that could not be confidently assigned to a specific habit. Additionally, we introduced an *aggregate* class to account for particles composed of multiple crystals, even though it does not represent a single-crystal habit in the traditional sense. To account for measurement artefacts, a separate *shattered* class was introduced for particles identified as fragments from shattering events; these particles were excluded from further analysis (see Sections 2.2.2 and 2.2.3). Representative stereo-images for each habit class are provided in the supplementary material.

### 2.2.2 Calculation of particle size distribution and bulk microphysical properties

Although the stereo-microscopic imager is limited to a maximum acquisition rate of 10 Hz, the trigger system of the PHIPS instrument can detect particles at a maximum acquisition rate of 13 kHz. The sensitive area of the instrument is determined by the overlap of the trigger system field of view and the scattering laser intensity distribution and varies between approximately 0.3 and 0.9 $\mathrm{mm}^2$, depending on the size of the particle (Waitz et al., 2021). This size-dependent sensitive area together with the

aircraft's true air speed is used to calculate the sampling volume of PHIPS. Due to the high velocity of a jet aircraft, compression of air occurs near probe and wing edges. To obtain the true ambient sampling volume, a thermodynamic correction described by Weigel et al. (2016) is applied.

The sizing of individual ice crystals is primarily based on the stereo-microscopic images. The images of the same particle from two viewing angles are used to derive the mean $D_{eq}$ and $A_{proj}$ per particle. In case a particle trigger event did not

generate a stereo-image due to the limitations of the imaging rate, $D_{eq}$ and $A_{proj}$ were derived based on the partial scattering cross section measured by the nephelometer as discussed in Waitz et al. (2021). During CIRRUS-HL flights, typically 70-80% of the particle triggers also produced stereo-images, with individual cases ranging from 10% to over 90%.



For the calculation of particle size distributions, particles were sorted into logarithmically spaced bins between 15 and 700 μm in $D_{eq}$. Two types of particle size distributions (PSDs) were genrated: one standard PSD including all detected particles (i.e., all trigger events), and another including only the subset of imaged particles that were manually classified as intact (i.e., not affected by shattering). For the standard PSD, potential shattering events were filtered by applying an interarrival time threshold of 0.1 ms. The shattering-corrected dataset was used to relate microphysical properties to the simultaneously retrieved angular light scattering properties which were likewise derived only for manually classified intact particles (see Sec. 2.2.3). For both PSD types, particles were binned into logarithmically spaced $D_{eq}$ intervals, and number concentrations were computed by dividing the number of particles in each bin by the corresponding sampling volume. The bulk microphysical properties ice water content (IWC), extinction coefficient ($\beta_{ext}$), effective radius ($r_{eff}$), and mass-equivalent spherical radius ($r_{\bar{m}}$) were derived using the particle-resolved size and area information.

The extinction coefficient ($\beta_{ext}$) was calculated based on the $A_{proj}$ of individual particles, measured in discrete size bins. For each size bin $i$, the projected area concentration $dA_{proj,i}$ was determined as

$$dA_{proj,i} = \frac{\sum_k A_{proj,i,k}}{V_i},\tag{1}$$

where $A_{proj,i,k}$ is the projected area of the $k$-th particle in bin $i$, and $V_i$ is the corresponding sampling volume for that bin. The summation is performed over all detected particles $k$ within the bin.

The total extinction coefficient was then obtained by summing over all size bins:

$$b_{ext} = 2 \cdot \sum_i dA_{proj,i},\tag{2}$$

where the factor of 2 assumes randomly oriented particles and geometric optics (Bohren and Huffman, 1998).

The IWC was calculated in a similar way by summing the masses of individual particles detected within each size bin. The mass $m_{i,k}$ of each particle $k$ in bin $i$ was estimated from its projected area $A_{proj,i,k}$ using the empirical power-law relationship from Baker and Lawson (2006):

$$m_{i,k} = 0.115 \cdot A_{proj,i,k}^{1.218},\tag{3}$$

where $A_{proj,i,k}$ is the projected area of each particle $k$ in bin $i$.

The mass concentration per bin $dm_i$ was then calculated as:

$$dm_i = \frac{\sum_k m_{i,k}}{V_i},\tag{4}$$

where $V_i$ is the bin-specific sampling volume.

The total IWC was obtained by summing over all size bins:

$$\text{IWC} = \sum_i dm_i.\tag{5}$$

The effective radius $r_{eff}$ was derived using:

$$r_{eff} = \frac{3}{2} \cdot \frac{\text{IWC}}{\rho_{ice} \cdot \beta_{ext}},\tag{6}$$



where $\rho_{\mathrm{ice}}$ is the bulk density of ice.

The mass-equivalent spherical radius $r_{\bar{m}}$ was calculated as:

$$r_{\bar{m}} = \left( \frac{3\,\mathrm{IWC}}{4\pi\,N\,\rho_{\mathrm{ice}}} \right)^{1/3}, \tag{7}$$

where $N$ is the total number concentration. The mass-equivalent sphere radius $r_{\bar{m}}$ represents the radius of a sphere with the average mass per particle of the ensemble, and is useful for linking two-moment microphysical schemes to bulk optical properties (Baran et al., 2025).

Figure 2 presents the mean particle size and projected area distributions for mid-latitude and Arctic cirrus cases, averaged over the entire CIRRUS-HL campaign. Distributions are shown for both all detected particle triggers and the subset of manually classified intact particles. The most notable differences between the two PSDs occur for small ice crystals ($D_{eq} < 100\,\mu\mathrm{m}$). In the standard PSDs, which use an interarrival time threshold to filter out shattering artefacts, the number concentration increases toward smaller sizes. However, this method does not fully eliminate shattering events, since shattering fragments can be detected individually and may not be distinguishable based on their timing alone. In contrast, the manually cleaned PSDs—based only on intact, imaged particles—show a decrease in number concentration toward smaller sizes for both regions, indicating more reliable removal of shattering artefacts. The discrepancy between the two PSDs is more pronounced in the mid-latitude cases, where larger ice crystals were more frequent and more prone to shattering upon impact.

This difference in the representation of small crystals between the two PSDs has implications for derived microphysical parameters. The $r_{\mathrm{eff}}$, which is weighted by particle area and extinction, is relatively insensitive to the presence of small shattered fragments. As a result, $r_{\mathrm{eff}}$ shows only minor differences between the standard and manually cleaned datasets (see Figure S2 in Supplementary Information). In contrast, $r_{\bar{m}}$ depends directly on both the number concentration and the IWC, making it more sensitive to spurious increases in small particle counts due to shattering. In the standard PSDs, this leads to an underestimation of $r_{\bar{m}}$. The manually cleaned dataset therefore provides a more physically representative estimate of $r_{\bar{m}}$, reinforcing the importance of careful shattering correction when interpreting airborne in situ measurements.

### 2.2.3 Retrieval of ice asymmetry parameter

The $g$ was retrieved from the ensemble-averaged angular scattering functions measured by the PHIPS instrument, following the method described in Xu et al. (2022). As with any polar nephelometer, the limited angular detection range of PHIPS necessitates a retrieval algorithm to accurately estimate the full scattering phase function. The retrieval approach is based on the assumption that the total asymmetry parameter in the geometric optics regime can be separated into contributions from geometrical optics and diffraction (Macke et al., 1996b).

The diffraction component, which is strongly peaked in the forward direction, largely falls outside the PHIPS measurement range and can therefore be estimated based on the measured (imaged) particle size using a parameterised function derived from scalar diffraction theory. The geometrical optics contribution, on the other hand, is retrieved from the angular scattering data by fitting Legendre polynomials to the PHIPS measurements. The asymmetry parameter is the mean of these two components.



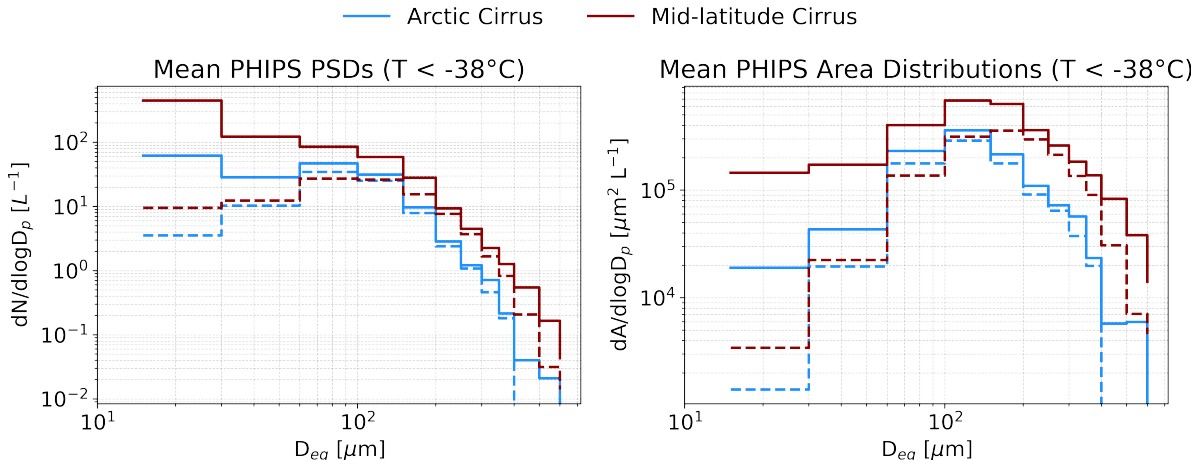

**Figure 2.** The mean PHIPS particle size distribution (PSD) and projected area size distribution for mid-latitude and Arctic cirrus. The solid lines show the mean based on 1-Hz PHIPS data in cirrus conditions (T < -38°C and IWC > $10^{-5}$ g m$^{-3}$) and the dashed lines the mean for only crystals with stereo-images that were manually classified as intact particles.

## 3 Results

To analyse cirrus microphysical and optical properties by geographical region, the CIRRUS-HL observations were classified into mid-latitude and Arctic cirrus using a latitude threshold of 65°N. Cirrus clouds were defined based on a combination of in-cloud temperature below -38°C and IWC greater than $10^{-5}$ g m$^{-3}$, derived from the 1 Hz PHIPS data. The campaign covered a wide geographical range, from 38.2°N to 75.6°N, with the highest cirrus cloud sampled at 14.5 km altitude and the coldest cirrus temperature recorded at -62.8°C. In total, approximately 5625 km of mid-latitude cirrus and 1060 km of Arctic cirrus were sampled, reflecting the greater spatial extent and coverage of mid-latitude observations during the campaign.

### 3.1 Ice crystal habits in mid-latitudes and Arctic cirrus

Figure 3 shows the fraction of habits as a function of $D_{eq}$ for the mid-latitude and Arctic cirrus cases. The most frequently occurring ice crystal habit class in both regions was the "other" category, accounting for 45% and 66% of all classified particles in mid-latitudes and Arctic, respectively. The high occurrence is especially pronounced for crystals smaller than 100 μm, for which over half of the measured particles fell into this group. In such cases, the classification to the "other" category can be due to the limited optical resolution available for morphological identification. Beyond this technical limitation, the "other" category also reflects the inherent complexity and variability of cirrus ice crystals, which often do not conform to traditional habit classifications (see Fig. 4 for example PHIPS images).

Irregular and compact ice crystals, broadly corresponding to our "other" category, have frequently been reported as the dominant habit type in both mid-latitude and Arctic cirrus (Garrett et al., 2001; Lawson et al., 2019; Wolf et al., 2018). For





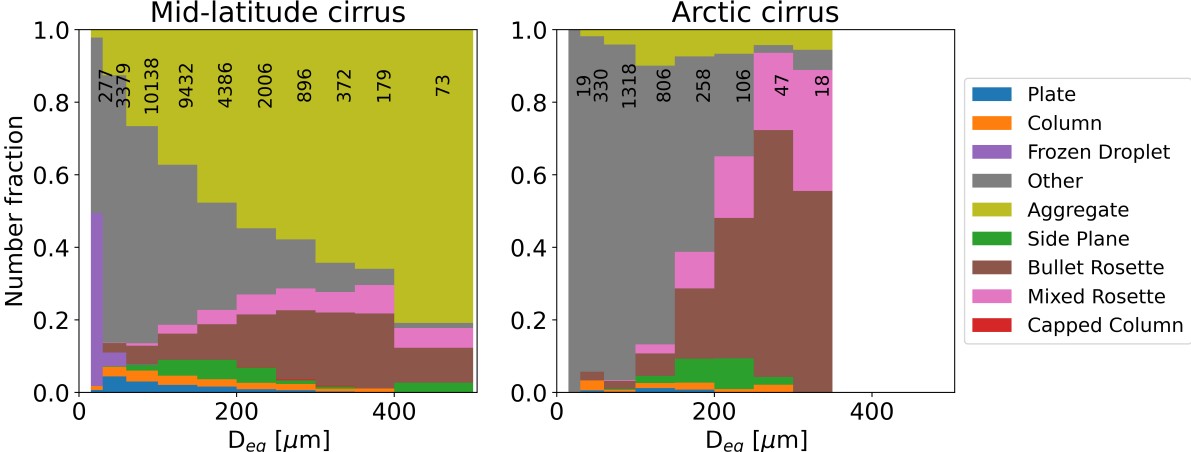

**Figure 3.** Ice crystal habit distribution as a function of the area equivalent diameter ($D_{eq}$) for mid-latitude and Arctic cirrus. The numbers indicate the number of ice crystal stereo-image pairs included in the habit analysis per size bin.

example, Wolf et al. (2018) observed that compact or irregular habits accounted for 50% and 90% of the crystal populations in Arctic liquid-origin and in-situ cirrus, respectively. In addition to observations from high-resolution imaging instruments, scanning electron microscope analyses have also confirmed the dominance of irregular shapes in cirrus clouds, even among
195 small ice crystals (Magee et al., 2021).

The most notable difference between mid-latitude and Arctic cirrus was a more frequent occurrence of aggregation in mid-latitudes compared to the Arctic. For mid-latitude cirrus aggregates was the second most frequent habit class, accounting for 32% of all classified crystals, whereas in Arctic cirrus they accounted only for 7% of all classified crystals. In mid-latitudes the fraction of aggregates was observed to clearly increase towards larger sizes with the highest fraction of 81% found for
200 the largest size group included in the analysis, between 400 and 500 µm. The more frequent occurrence of aggregates in mid-latitude cirrus can be explained through the more dynamic atmosphere at the lower latitudes (Barahona et al., 2017) and higher initial ice number concentration (Sourdeval et al., 2018), potentially promoting aggregation.

In mid-latitude cirrus most of the aggregates were consisting of plates (34%), suggesting a convective origin, or of crystals classified as "other" (27%). A large fraction of aggregates consisted of frozen droplets (25%) but these, together with the
205 simultaneously observed single frozen droplets, originated from research flight 12 targeting a strong overshooting convective system (see Schäfler and Rautenhaus, 2023). In contrast, frozen droplets and their aggregates were rarely detected in other mid-latitude cases, indicating that their occurrence was closely linked to specific convective conditions.

The second notable difference between the two regions was a higher occurrence of bullet and mixed rosettes in the Arctic. Bullet and mixed rosettes comprised 19% of all classified crystals and dominated the population larger than 200 µm, comprising
210 56–89% in that size range. This makes them the second most frequent habit type in Arctic cirrus. A similar dominance of bullet rosettes as the second most common habit was also reported in Arctic conditions by Wolf et al. (2018). For mid-latitude cirrus





**Figure 4.** Example ice crystal microscopic images recorded by PHIPS from mid-latitude and Arctic cirrus as a function of their area equivalent diameter ($D_{eq}$). The distribution of the different habits corresponds to Figure 3.




bullet and mixed rosettes were common but less frequent comprising 11% of all crystals and reaching maximum occurrence frequencies of 25–28% in the size range of 250–400 μm.

Other identified habits in both regions, each contributing between 2 and 9% to the total population, included side planes, as well as single columns. In contrast, single plates were rare in Arctic cirrus (0.5%) compared to 3.5% in mid-latitude cirrus.

## 3.2 Cirrus microphysical and optical properties in mid-latitudes and Arctic

Figure 5 presents the frequency distribution of key bulk microphysical properties of cirrus – ice number concentration, IWC, and $r_{\text{eff}}$ – along with $g$ as a function of latitude, derived from PHIPS observations for the size range of 15 to 700 μm. The selected microphysical parameters are commonly used as input for radiative transfer schemes and were derived from 10-second averaged PHIPS particle size distributions. The asymmetry parameter was calculated from populations of 50 consecutive ice crystals and includes only manually inspected and classified intact (non-shattered) particles. Similarly, $r_{\text{eff}}$ was derived using only non-shattered, imaged particles to ensure accurate representation of individual crystal morphology. In contrast, ice number concentration and IWC were calculated using the full dataset, including particles without images, since applying the same shattering removal would lead to a systematic underestimation of these quantities. The minimum detection limits for the 10-second averages were approximately 0.5 $\text{L}^{-1}$ for number concentration and 0.05 $\text{mg m}^{-3}$ for IWC, based on the instrument's sampling volume and estimated particle mass.

The median total concentration of ice crystals in the 15 to 700 μm size range varied between 1 and 52 $\text{L}^{-1}$ and showed no clear latitudinal trend. However, slightly higher overall median values were observed in mid-latitude cirrus (11 $\text{L}^{-1}$) compared to Arctic cirrus (8 $\text{L}^{-1}$) (Table 1). Mean values differed more noticeably, with 100 $\text{L}^{-1}$ in the mid-latitudes and 23 $\text{L}^{-1}$ in the Arctic. This difference was primarily driven by occasional high concentrations exceeding 100 $\text{L}^{-1}$ in the mid-latitude dataset, which were not observed in the Arctic. However, these elevated concentrations are likely influenced by particle shattering effects.

Similar to the total ice number concentration, the $IWC$ values did not exhibit a significant latitudinal trend between 42°N and 75°N, where most observations were conducted. Median $IWC$ values ranged from 0.2 to 10 $\text{mg m}^{-3}$, with an overall median of 2.8 $\text{mg m}^{-3}$ and 3.0 $\text{mg m}^{-3}$ for mid-latitudes and Arctic cirrus, respectively. Also the mean values were similar for mid-latitude and Arctic cirrus, 2.6 and 2.4 $\text{mg m}^{-3}$, respectively, which confirms that the higher number concentrations observed in the mid-latitudes are from small ice crystals that do not contribute significantly to the $IWC$.

The only microphysical parameter that shows a clear latitudinal dependence was $r_{\text{eff}}$, with median values increasing from approximately 24 μm within latitudes from 39°N to 42°N to 45 μm within latitudes above 70°N. An exception to this trend was observed near 62°N, where median effective radii around 50 μm were recorded in a liquid-origin system sampled off the coast of Norway during the double flight F18/F19. A similar trend towards higher $r_{\text{eff}}$ values in more pristine environments was observed by Gayet et al. (2004) during the Interhemispheric Differences in Cirrus Properties From Anthropogenic Emissions (INCA) field experiments, where the average northern hemispheric effective radius (36 μm) was smaller than that observed in the southern hemisphere (42 μm).



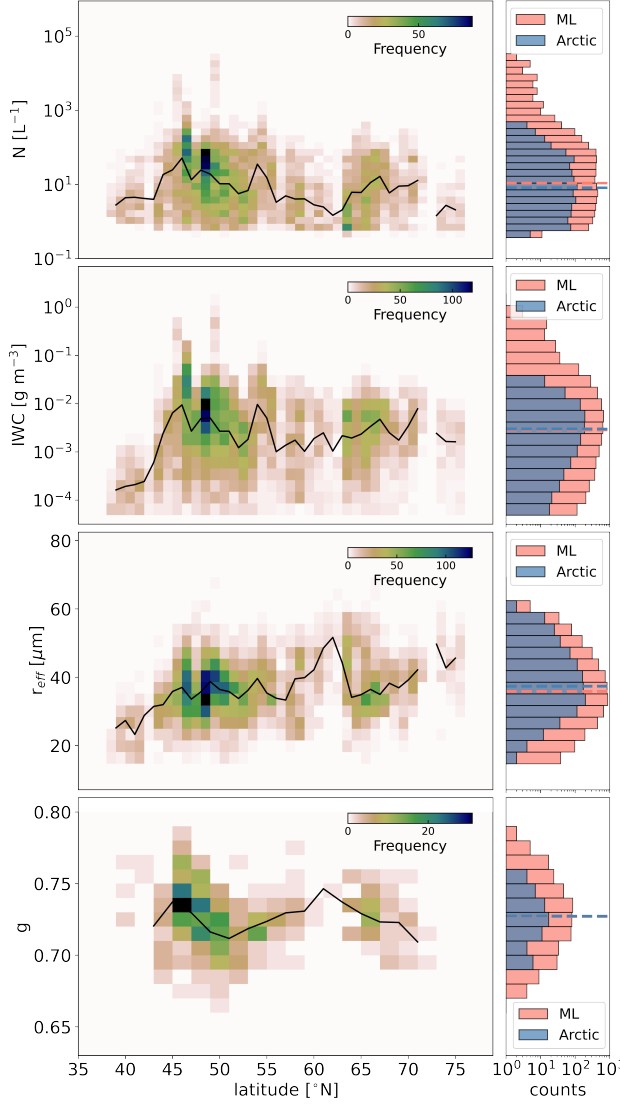

**Figure 5.** Frequency distribution of total concentration ($N$), effective radius ($r_{\text{eff}}$), ice water content ($IWC$) and asymmetry parameter ($g$) observations from the PHIPS as a function of latitude. Microphysical observations are 10-second average values and asymmetry parameter is retrieved for only particles with stereo-images where shattering artefacts are excluded. The black solid lines mark the median line. Latitude bin widths are 1° for microphysical observations and 2° for the asymmetry parameter observations. The histograms show the distribution of the values for all mid-latitude (ML) and Arctic cases using a latitude threshold of 65°N. The dashed line over the histograms indicate the median value.

In contrast to the trend in $r_{\text{eff}}$, the asymmetry parameter $g$ showed no systematic latitudinal variation, with median values ranging from 0.710 to 0.745. Notably, the median asymmetry parameter was identical in both regions (0.727) indicating a remarkable similarity in the bulk optical scattering characteristics.





**Table 1.** Summary statistics (median, mean, and percentiles) of microphysical and optical properties derived from PHIPS measurements in mid-latitude (ML) and Arctic cirrus clouds. Shown are total number concentration ($N$), ice water content ($IWC$), effective radius ($r_{\mathrm{eff}}$), and asymmetry parameter ($g$). The analysis includes particles in the PHIPS measurement range of 15 to 700 μm.

| Variable | Statistic | ML | Arctic |
|---|---|---|---|
| N [L$^{-1}$][1] | Median | 10.9 | 8.2 |
| | Mean | 100.1 | 22.7 |
| | 25th percentile | 3.0 | 2.5 |
| | 75th percentile | 45.4 | 25.7 |
| | 5th percentile | 0.9 | 0.8 |
| | 95th percentile | 187.3 | 83.3 |
| IWC [mg m$^{-3}$][1] | Median | 2.8 | 3.0 |
| | Mean | 2.6 | 2.4 |
| | 25th percentile | 0.7 | 1.1 |
| | 75th percentile | 8.7 | 6.2 |
| | 5th percentile | 0.13 | 0.19 |
| | 95th percentile | 35.5 | 14.1 |
| r$_{eff}$ [$\mu$m] | Median | 35.9 | 37.4 |
| | Mean | 36.1 | 38.2 |
| | 25th percentile | 30.9 | 33.1 |
| | 75th percentile | 41.1 | 42.5 |
| | 5th percentile | 23.8 | 27.8 |
| | 95th percentile | 49.2 | 51.2 |
| $g$ | Median | 0.727 | 0.727 |
| | Mean | 0.726 | 0.725 |
| | 25th percentile | 0.714 | 0.715 |
| | 75th percentile | 0.738 | 0.734 |
| | 5th percentile | 0.694 | 0.698 |
| | 95th percentile | 0.761 | 0.751 |

[1]Lower detection limits for 10-second averages are approximately 0.5 L$^{-1}$ for $N$ and 0.05 mg m$^{-3}$ for IWC.

The mid-latitude cases covered a broader range of cirrus conditions, while Arctic sampling was limited to five predominantly liquid-origin cirrus systems north of 65°N. Due to pandemic-related base relocation to Germany, Arctic flights required long transits and thus targeted synoptic cirrus with predictable development: a warm frontal cirrus (F05), two liquid-origin systems (F08/F09), a jet-stream-associated liquid-origin cirrus (F10/F11), and a mix of liquid-origin and in-situ cirrus over Iceland (F13/F14, Fig. 1). This focus introduces a bias toward liquid-origin cirrus, which have been shown to exhibit higher $IWC$





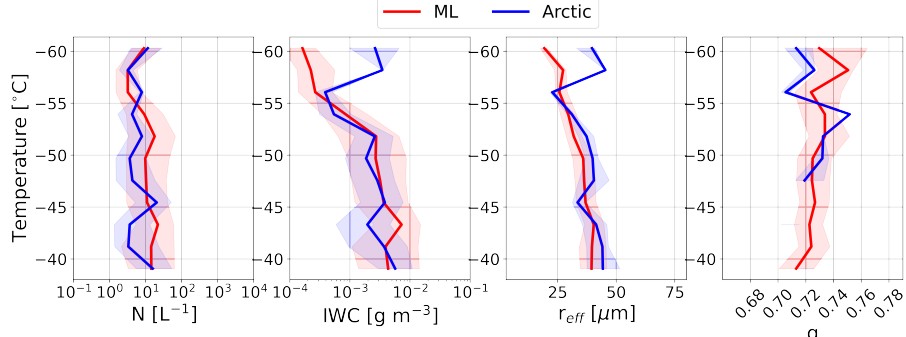

**Figure 6.** Statistical analysis of the observed total number concentration ($N$), ice water content ($IWC$), effective radius ($r_{\text{eff}}$), and asymmetry parameter ($g$) as a function of in-cloud temperature in degree Celsius. The solid line shows the median values and the shaded area the interquartile range for mid-latitude (ML; red) and Arctic (blue) observations using a latitude threshold of $65°$.

compared to in situ-formed cirrus (Luebke et al., 2016). This sampling bias should be taken into account when interpreting observed latitudinal trends in cirrus microphysical and optical properties.

To better account for thermodynamic influences on cirrus properties, it is useful to examine the observations as a function of in-cloud temperature rather than geographic location. Figure 6 shows the statistical distribution of the ice number concentration, $IWC$, $r_{\text{eff}}$ and $g$, comparing mid-latitude (red) and Arctic (blue) observations. Median ice number concentrations tend to be higher in the mid-latitudes for temperatures down to approximately $–55°$C. Below this threshold, Arctic values become comparable to, or even exceed, those in the mid-latitudes. However, the observed differences are only statistically significant across several temperature intervals between $–42°$C and $–55°$C (two-sample t-test, $p<0.05$).

$IWC$ was generally similar in both regions at temperatures below $–55°$C, but became higher in Arctic cirrus at colder temperatures. However, due to the limited number of observations this difference is not statistically significant ($p>0.05$). The $r_{\text{eff}}$ was mostly larger in the Arctic compared to the mid-latitudes with median values exceeding those in the mid-latitudes by $5\,\mu$m and showing a decreasing trend with temperature for temperatures down to $–55°$C. At temperatures colder than $–55°$C, the $r_{\text{eff}}$ of Arctic cirrus increases again, showing up to twice as large $r_{\text{eff}}$. The observed differences were found to be statistically significant in the temperature ranges between $-38°$C and $-40°$C, and between $-44°$C and $-52°$C. In contrast, the asymmetry parameter $g$ showed only a weak temperature dependence. Slightly lower values were observed in Arctic cirrus at temperatures below $–55°$C, but as with the other parameters, the observed differences at colder temperatures are not statistically significant.

Inspection of Table 1 shows that the microphysical properties derived from the PHIPS instrument are in general agreement

with previously reported values in the mid-latitudes and Arctic (Luebke et al., 2016; Heymsfield et al., 2017; Wolf et al., 2018). The median values of the total number concentration are systematically higher than those reported in the literature based on optical array probes with larger sampling volumes (Luebke et al., 2016; Heymsfield et al., 2017; De La Torre Castro et al., 2023). This discrepancy arises because low concentrations, below approximately $0.5\,\text{L}^{-1}$, are not detected with PHIPS within the 10-second averaging period. Values of $N$ and $IWC$ obtained from optical array probe measurements during the campaign





**Table 2.** Pearson correlation coefficients ($r$) and p-values for the relationship between $g$ and various atmospheric and microphysical parameters. Moderate correlations are highlighted.

| Variable | $r$ | p-value |
|---|---|---|
| Temperature | -0.19 | 0.000 |
| $RH_i$ | -0.11 | 0.016 |
| Aspect ratio | -0.12 | 0.010 |
| $D_{eq}$ | **-0.41** | 0.000 |
| log(IWC) | -0.08 | 0.100 |
| $\mathbf{r}_{eff}$ | **-0.28** | 0.000 |
| $r_{\bar{m}}$ | **-0.32** | 0.000 |
| N | -0.16 | 0.000 |

are discussed in detail by De La Torre Castro et al. (2023) and Jurkat-Witschas et al. (2025). Furthermore, the observed effective radii ($r_{\text{eff}}$) in both regions are significantly lower than the values reported by De La Torre Castro et al. (2023) for the same study. This discrepancy is most likely due to their assumption of a simplified spherical ice particle shape to derive this quantity from the measured particle size distribution.

### 3.3 Sensitivity of g to ice microphysical and environmental factors

To investigate the correlation between $g$ and simultaneously measured in-cloud temperature, relative humidity over ice ($RH_i$), and cloud microphysical properties, these variables were first averaged over the same time intervals that were used for calculating $g$ (i.e. time intervals that contain 50 consecutive intact ice crystals). These averaging periods ranged from 44 seconds to 4 minutes (25th to 75th percentile), corresponding to sampled cloud volumes between approximately 3 and 20 L. The microphysical parameters include the mean $D_{eq}$ and $AR$ of the same 50 consecutive ice crystals. Additionally, properties derived

from the PHIPS particle size distributions, such as $IWC$, $r_{\text{eff}}$, and number concentration ($N$), were analysed according to Section 2.2.2. To ensure consistency, we derived these properties from the manually cleaned data set - also for the number concentration and $IWC$.

In our analysis, we also included the mass-equivalent spherical radius, $r_{\bar{m}}$, as defined by Baran et al. (2025). This parameter represents the radius of a sphere with the mean mass of the ice crystal population and has been proposed as a more physically

consistent quantity for linking two-moment microphysical schemes with optical properties. In contrast to the effective radius $r_{\text{eff}}$, $r_{\bar{m}}$ does not require assumptions about the ice particle surface area.

Table 2 summarises the Pearson correlation coefficients between $g$ and the investigated parameters. Among the tested parameters, the strongest correlation was observed with $D_{eq}$ ($r = -0.41$), followed by $r_{\bar{m}}$ ($r = -0.32$) and $r_{\text{eff}}$ ($r = -0.28$), all indicating moderate negative correlations. These results suggest that $g$ decreases with increasing particle size. Only weak

correlations were found with temperature, $RH_i$, AR, and number concentration ($N$).




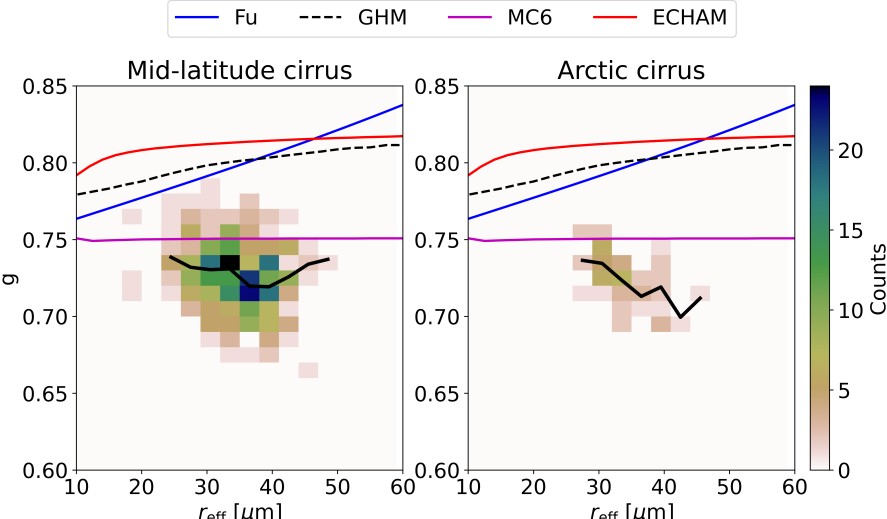

**Figure 7.** Frequency distribution of asymmetry parameter ($g$) observations as a function of effective radius ($r_\text{eff}$). Both the asymmetry parameter and effective radius are retrieved only for particles with stereo-microscopic image that were not classified as shattering. The black line indicated the median value for the effective radius bins with a bin width of 3 μm. Existing short wave optical parameterisations are overlaid to the measurement data. These are the parameterisation by Fu (1996) (Fu; blue line), general habit mixture (GHM, dashed black line; Cole et al., 2013), the severely roughened column aggregate model used in MODIS Collection 6 retrievals (MC6; Platnick et al., 2016) and the optical parameterisation used in ECHAM6 general circulation model (ECHAM; Pincus and Stevens, 2013).

Figure 7 shows the frequency distribution of the measured $g$ as a function of $r_\text{eff}$ for ML and Arctic cirrus cases. The same figure for $r_{\bar{m}}$ is shown in Supplementary Information (Fig. S11). It can be seen that $g$ values vary between 0.65 and 0.79. In the ML dataset, the median $g$ decreases from 0.74 to 0.72 with increasing $r_\text{eff}$ up to approximately $r_\text{eff}$=40 μm, beyond which the trend reverses and the median $g$ increases with particle size. However, this upward trend at larger sizes is not statistically robust due to the limited number of observations in that range. In Arctic cirrus, the data are more sparse overall, but a consistent decrease in the median $g$ from 0.74 to 0.70 with increasing $r_\text{eff}$ is observed across the measured range.

In radiative transfer simulations, ice optical properties are frequently parameterised as a function of $r_\text{eff}$. Figure 7 shows four commonly used parameterisations of $g$ at a wavelength band centered at 532 nm, corresponding to the PHIPS light scattering measurements. The first parameterisation, developed by Fu (1996), is based on columnar crystals with single scattering properties calculated using an improved geometric optics method. The asymmetry parameter is expressed as a function of a generalised effective size, which can be related to $r_\text{eff}$ via Eq. 3.12 in Fu (1996). The second parameterisation represents the General Habit Mixture (GHM), which assumes a combination of nine crystal habits with severe surface roughness ($\sigma = 0.5$) (Cole et al., 2013). The third is the MODIS Collection 6 (MC6) parameterisation, which assumes aggregates composed of severely roughened columns (Platnick et al., 2016). The fourth is the parameterisation implemented in the ECHAM6 general circulation model (Pincus and Stevens, 2013).





Among the four, only the MC6 parameterisation—assuming complex, severely roughened aggregates with a fixed asymmetry parameter of 0.75—captures the upper range of the observed values, but not the median or the lower range. The remaining parameterisations overestimate $g$ across the entire size range and fail to reproduce any of the measured values. Moreover, all four parameterisations predict an increasing trend in the $g$ with increasing $r_{\text{eff}}$, which contrasts with the observational data showing a clear decreasing trend.

### 3.3.1 Analysis based on individual particle classes

A key advantage of PHIPS as a single particle polar nephelometer is its ability to directly link the microphysical and optical properties of individual ice crystals. This allows for the selection of particles with specific microphysical characteristics, such as size or aspect ratio, and enables the construction of "artificial" particle populations consisting of crystals with similar properties. This approach results in a very detailed investigation of how individual microphysical parameters influence $g$, isolated from the natural microphysical variability present in real cloud segments. This approach was recently used to retrieve the asymmetry parameter of bullet rosettes from PHIPS data captured during CIRRUS-HL (Wagner et al., 2024).

Figure 8 presents a statistical analysis of $g$ for all ice crystals measured during CIRRUS-HL that were manually classified as intact. The ice crystals were grouped based on their $D_{eq}$ (for bin definitions, see Waitz et al., 2021), $AR$ (defined as the smaller of the two aspect ratios from the stereo images) using a bin width of 0.1, and simultaneously measured ice supersaturation ($S_{\text{ice}}$), also using a bin width of 0.1. Each bin contains a minimum of 100 ice crystals. To retrieve $g$, the crystals within each bin were further divided into groups of 50 consecutive particles, and one $g$ was calculated per group. The number of these groups is indicated in the figure. Additionally, a single $g$ was retrieved for all crystals within each bin, with the corresponding value and uncertainty shown as markers.

Of all the variables, only $D_{eq}$ shows a clear decreasing trend with $g$. This is consistent with the results from the previous analysis, where cloud volumes were analysed instead of particle classes. The highest median values ($g = 0.763$), were measured for mid-latitude ice crystals with $D_{eq}$ between 15 and 30 µm, while the lowest median values ($g = 0.710$) were observed for $D_{eq}$ between 150 and 200 µm. For larger particles, the $g$ increases again, although this trend is not statistically robust due to fewer observations.

This increase can likely be explained by the influence of ice crystal habit to $g$. Figure 9 illustrates the variability of $g$ across different ice crystal habits and size ranges, providing further insight into the influence of habit on the observed size dependence of g. The analysis was conducted for the seven most frequently occurring habit categories: aggregate, column, side plane, frozen droplet, bullet rosette, plate, and the "other" category. For each habit–size combination, a median g was retrieved for each size bin, provided that at least 50 intact crystals were available in the bin.

The general size dependence of $g$ observed in Figure 8 is seen across most habits, except for frozen droplets that only cover the two smallest size bins. A maximum difference of 0.05 in $g$ is found between columns, which show the lowest values, and aggregates, which exhibit the highest values. This difference exceeds the measurement uncertainty and is therefore considered to be significant. Side planes, bullet rosettes, plates, and crystals in the "other" category all show similar $g$ values within the





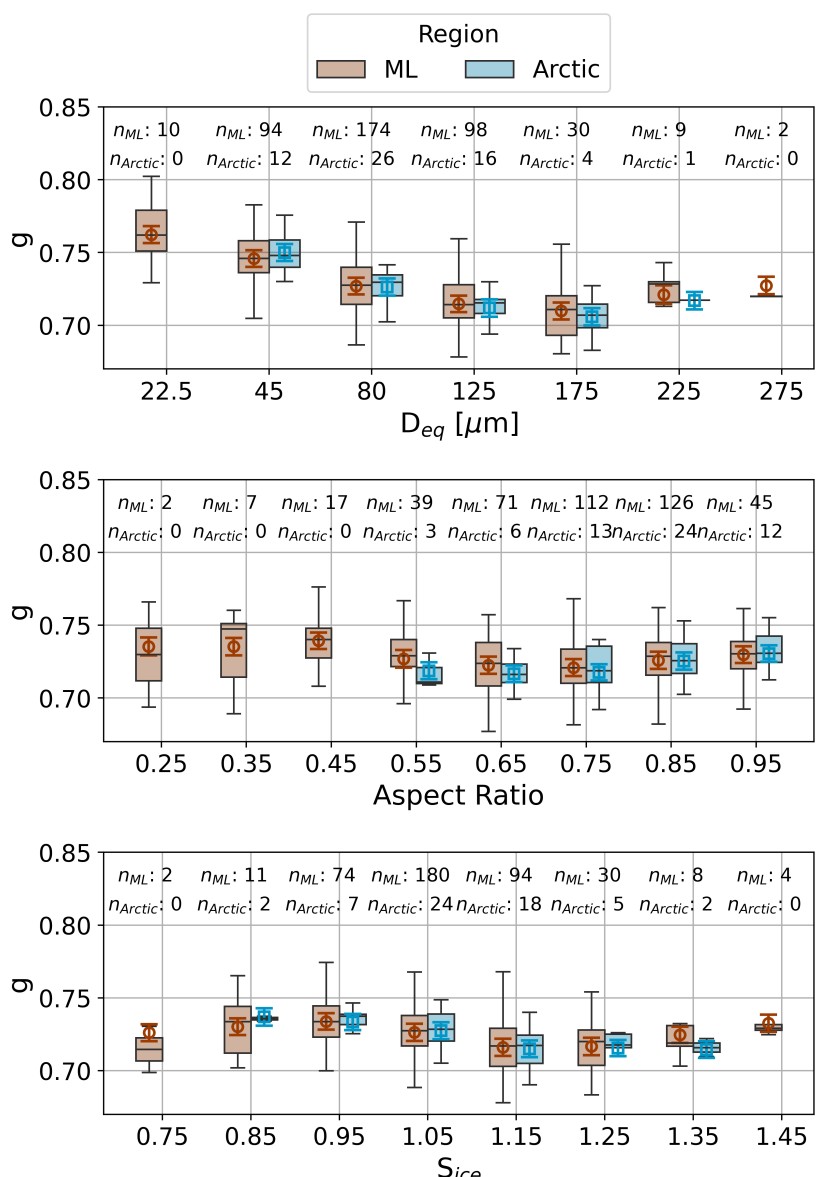

**Figure 8.** Statistical analysis of the asymmetry parameter ($g$) for mid-latitude (ML) and Arctic cirrus as a function of (a) particle area equivalent diameter ($D_{eq}$), (b) aspect ratio, and (c) ice supersaturation ($S_{ice}$). The box indicates the 25th and 75th percentiles (quartiles), the line the 50th percentile (median) and the whiskers the 5th and 95th percentiles of the $g$ values retrieved for samples of 50 crystals. The markers indicate the asymmetry parameter retrieval for all crystals within the bin with error bars denoting the retrieval uncertainty. Sample sizes corresponding the number of asymmetry parameter retrievals for groups of 50 particles for each category are annotated in the plots.

uncertainty range for sizes below 200 μm. For larger bullet rosettes, $g$ increases and becomes comparable to that of aggregates
of similar size.





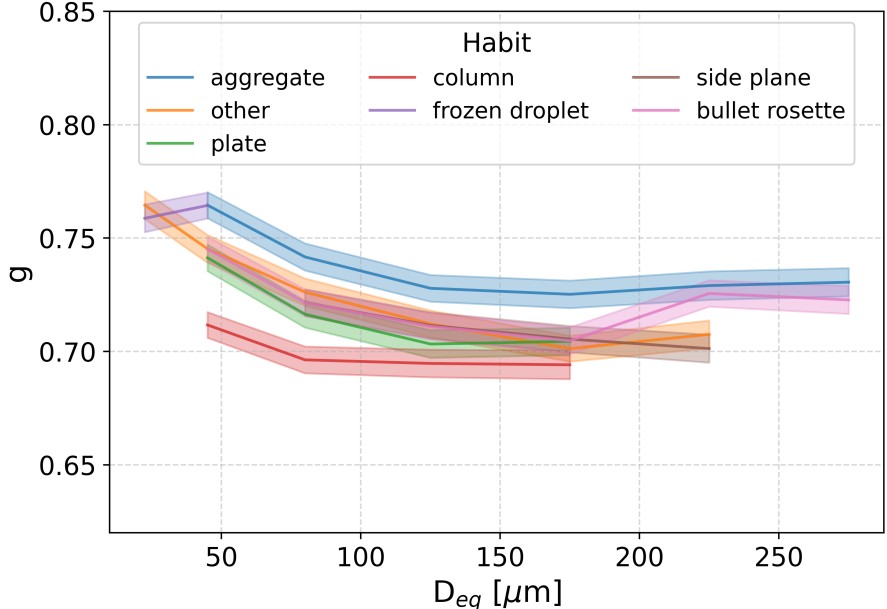

**Figure 9.** Mean asymmetry parameter, $g$, as a function of particle diameter for various ice crystal habits, including aggregates, plates, columns, frozen droplets, side planes, and bullet rosettes. The shaded areas represent the retrieval uncertainty in asymmetry parameter for each habit.

No significant difference was found in the median $g$ between mid-latitude and Arctic cirrus with median $g$ approximately between 0.72 and 0.74 observed for each $AR$ and $S_{ice}$ category. In general, the variability in $g$ was greater in the mid-latitudes across all categories (g varied between 0.667 and 0.802 for groups of 50 crystals) compared to the Arctic (g varied between 0.683 and 0.776). However, this may be partly attributed to the smaller sample size in the Arctic.

## 4 Discussion

Despite differences in geographic location and the ice crystal habit distributions, the $g$ values showed no significant variation between mid-latitude and Arctic cirrus but consistently low median values were observed. Low $g$ values below 0.8 have been systematically measured also by other polar nephelometers for the visible spectral range. Measurements with the airborne CIN instrument have shown $g$ values around 0.74±0.03 for Arctic ice clouds (Gerber et al., 2000; Garrett et al., 2001) and values around 0.75±0.01 for tropical anvil cirrus (Garrett et al., 2003). In mid-latitudes, median $g$ values between 0.769 and 0.79 have been measured using the PN instrument (Gayet et al., 2004; Febvre et al., 2009; Auriol et al., 2001; Gayet et al., 2012).

Our results show that $g$ is largely insensitive to variations in atmospheric conditions (temperature and relative humidity), as well as to most microphysical properties of ice crystals. This finding is consistent with several previous observational studies. For example, Febvre et al. (2009) reported similar optical properties for natural cirrus and aged contrails, despite differences





in their size distributions. Likewise, Garrett et al. (2001) found that $g$ values were insensitive to habit variations in Arctic cirrus. During the INCA experiment, similar $g$ values were derived for the northern and southern hemispheres (Gayet et al., 2004, 2006), with only a weak temperature dependence noted by Gayet et al. (2006).

Here, the only parameter for which a moderate correlation with $g$ was observed is particle size: we find that $g$ values tend to decrease with increasing $D_{eq}$, $r_{\mathrm{eff}}$ or $r_{\bar{m}}$. This behaviour contrasts with the findings of Gayet et al. (2004), who reported an

increasing trend in $g$ with effective diameter. In contrast, Garrett et al. (2003) found no clear dependence between $g$ and $r_{\mathrm{eff}}$ for small anvil ice crystals with $r_{\mathrm{eff}} < 20\,\mu\mathrm{m}$. Interestingly, the aspect ratio ($AR$) does not appear to influence $g$ in our data, which stands in contradiction to theoretical studies predicting a strong dependence between $g$ and $AR$, even for roughened crystals (Iaquinta et al., 1995; van Diedenhoven et al., 2014).

The low median $g$ value of 0.727 cannot be explained by pristine hexagonal shapes (Iaquinta et al., 1995; Schmitt et al.,

2006). Rather, a significant degree of surface roughness or other deformations are needed in order to numerically reproduce our observations (Macke et al., 1996a; Yang et al., 2013; Wagner et al., 2024). Given that the 95th percentiles of the $g$ distributions are 0.761 for mid-latitude and 0.751 for Arctic cirrus, we conclude that the majority of the measured ice crystals exhibit a significant degree of morphological complexity. These findings suggest that crystal complexity plays a more dominant role in determining $g$ than any other microphysical property. The observed size dependence of $g$ may also reflect an increasing degree

of crystal complexity in larger ice particles.

In-situ measurements have indicated $g$ values above 0.8 only for a few cases. Garrett et al. (2001) reported g values of 0.81 for sublimating ice crystals at $\mathrm{RH}_{ice}$ around 87%, and Gayet et al. (2012) found similar $g$ values for $\mathrm{RH}_{ice}$ below 80%. Febvre et al. (2009) observed $g$ values of 0.827 for young contrails, with a mean $\mathrm{RH}_{ice}$ of 118%. In our dataset, we did not observe an increase in $g$ at lower $\mathrm{RH}_{ice}$ values. However, our observations included only a limited number of ice crystals at $\mathrm{RH}_{ice}$ below

80%. Therefore, the effect of sublimation on $g$ remains inconclusive and needs further investigation.

A limitation of this study is that $g$ retrievals were restricted to ice crystals with $D_{eq}$ below 300 μm. This constraint arises because larger particles frequently saturated the first few measurement channels of our polar nephelometer, preventing a reliable retrieval of $g$. However, as shown in the particle size and area distributions in Figure 2 and the cumulative distributions in Figure S1, ice crystals smaller than 300 μm account for 97% (99%) of the total number and 83% (94%) of the total cross-sectional

area in mid-latitude (Arctic) cirrus. Based on this, we consider our retrieved $g$ values to be representative of the cirrus cloud populations sampled during the CIRRUS-HL campaign.

Our results show a substantial discrepancy between the parameterisations of $g$ used in climate models and the values measured in this study (see Fig. 7). This difference has significant implications for inferred cloud reflectivity or albedo, which can be demonstrated using a simple approximation of the plane albedo for thin cirrus clouds (Meador and Weaver, 1980)

$$R = \frac{\omega_0}{2\,\mu_0}\,(1-g)\,\tau, \tag{8}$$

where $\omega_0$ is the single-scattering albedo, $\mu_0$ is the cosine of the solar zenith angle, and $\tau$ is the optical thickness. For constant optical thickness and solar angle, the plane albedo is directly proportional to $1-g$. Therefore, a decrease of $g$ from 0.8 to the measured 0.727 corresponds to an increase in albedo of approximately 37%, making cirrus clouds more reflective at solar



wavelengths. This indicates that an overestimation of $g$ can lead to a significant bias in the estimation of the radiative effect of cirrus clouds. Additionally, such an overestimation can introduce large errors in the retrieval of cirrus optical depth from space-borne radiance measurements (Yang et al., 2008; Yi et al., 2017a).

Studies using general circulation models (GCMs) or Moderate Resolution Imaging Spectroradiometer (MODIS) retrievals to simulate top-of-the-atmosphere (TOA) fluxes have shown that incorporating optical parameterisations with lower $g$ values in the visible spectrum leads to a closer agreement between these TOA flux simulations and TOA flux estimates from direct measurements of radiances using space-borne remote sensing (Kristjánsson et al., 2000; Yi et al., 2017b). However, Ren et al. (2023) found that optical parameterisations with low $g$ values around 0.75 overestimated both shortwave (SW) and longwave (LW) fluxes at TOA compared to observations from the Clouds and the Earth's Radiant Energy System (CERES). Their study emphasised the need for further optimisation of the scattering phase function to achieve radiative closure. Yi et al. (2017b) also showed that lowering $g$ (by changing the optical parameterisation from MC5 to MC6) significantly reduced the estimated TOA SW flux from MODIS observations. This reduction resulted from lower cloud optical thickness retrieved by the MC6 model, which was subsequently used as input to the radiative transfer simulations (Yi et al., 2017a).

Currently, cirrus and ice containing ice clouds have been estimated to exert a globally and annually averaged net warming effect of 2-5 $\mathrm{Wm^{-2}}$ (Hong et al., 2016; Yi et al., 2017b; L'Ecuyer et al., 2019). Modelling studies using GCMs have shown that lowering the $g$ values in visible from values around 0.8 to 0.75 will decrease the net ice CRE by 1-2 $\mathrm{Wm^{-2}}$ (Järvinen et al., 2018; Yi et al., 2013; Yi, 2022). Our results suggest that an additional decrease of the $g$ values from 0.75 to values around 0.727 might be needed, which would lead to an even higher bias in CRE.

A potential bias in CRE estimates is highly significant in the context of the Earth's energy balance. The current estimate of the global radiative imbalance is on the order of 0.5 to 1 $\mathrm{Wm^{-2}}$ (Loeb et al., 2021), meaning that even small biases in cirrus CRE can mask or amplify the signals of anthropogenic forcing. Cirrus clouds cover up to 30% of the globe at any given time (Stubenrauch et al., 2013), and errors in their radiative properties can accumulate spatially and temporally, introducing systematic uncertainty into global climate models. A positive bias in cirrus CRE leads to an overestimation of their warming effect, potentially distorting the magnitude and sign of cloud feedbacks (Boucher et al., 2013). This can also affect the tuning of GCMs, where compensating adjustments in low cloud properties or surface fluxes may be introduced to maintain top-of-atmosphere energy closure (Mauritsen et al., 2012), at the expense of physical realism. Improving the representation of cirrus optical properties, particularly $g$, is therefore essential for accurately quantifying cloud–radiation interactions and reducing uncertainty in climate projections.

## 5 Summary

We analysed ice crystal microphysical and correlated angular light scattering properties of mid-latitude and Arctic cirrus clouds using in situ measurements performed with the PHIPS airborne cloud probe during the CIRRUS-HL campaign. Ice crystal number concentration and ice water content showed no clear latitudinal trends, with similar median values of 11 $\mathrm{L^{-1}}$ and 2.8 $\mathrm{mg\,m^{-3}}$ for mid-latitude cirrus and 8 $\mathrm{L^{-1}}$ and 3.0 $\mathrm{mg\,m^{-3}}$ for Arctic cirrus. In contrast, the effective radius increased



toward higher latitudes. Furthermore, clear regional differences in ice crystal habit distributions were observed: aggregates were the dominant habit in mid-latitude cirrus for crystals larger than 200 μm, while bullet and mixed rosettes dominated Arctic cirrus. Smaller crystals in both regions were primarily complex in shape and unclassifiable using standard habit categories.

Despite the differences in habits, both mid-latitude and Arctic cirrus exhibited nearly identical asymmetry parameter values, with the same median asymmetry parameter of 0.727 for both regions and observed values as low as 0.694. The asymmetry parameter was largely insensitive to environmental conditions such as temperature and relative humidity, as well as to ice crystal aspect ratio and bulk microphysical parameters including IWC and number concentration. Moderate correlations were found only with particle size.

The relatively low asymmetry parameter values cannot be represented using optical models for idealised pristine crystals. Comparison with four commonly used radiative transfer parameterisations revealed that most overestimated asymmetry parameter by up to 0.1. Only the MODIS Collection 6 parameterisation, which assumes severely roughened columnar crystals, captured the upper range of the observed values. Given the inverse relationship between the asymmetry parameter and cloud reflectivity, overestimating asymmetry parameter from 0.727 to 0.800 can lead to an underestimation of cirrus albedo by up to 440    37%, with implications for both radiative forcing estimates and satellite retrievals of optical depth. Such biases are particularly important because cirrus clouds have estimated to exert a net warming effect of 2–5 $\mathrm{W\,m^{-2}}$ globally, and reducing asymmetry parameter in climate models by just 0.05 has been shown to lower this warming by 1–2 $\mathrm{W\,m^{-2}}$. These discrepancies highlight the need for improved observational constraints and parameterisations of asymmetry parameter to better quantify the radiative role of cirrus clouds and reduce uncertainties in climate projections.

*Data availability.* Processed PHIPS microphysical data and single-particle light scattering data will be available in the HALO database (https://halo-db.pa.op.dlr.de/mission/125) upon publication. The stereo-microscopic images are available from the authors upon request.

*Author contributions.* MS designed the measurement concept of PHIPS and led the development of the instrument. EJ and MS collected the PHIPS data during the CIRRUS-HL campaign. EJ performed the data analysis and wrote the manuscript. MS contributed to the data interpretation and provided comments on the manuscript.

*Competing interests.* Martin Schnaiter and Emma Järvinen are affiliated with schnaiTEC GmbH, the manufacturer of the PHIPS instrument. Martin Schnaiter is additionally employed part-time by schnaiTEC GmbH.

*Acknowledgements.* The authors would like to thank the HALO crew for supporting the operations during the CIRRUS-HL campaign and the science and weather forecast teams for flight planning. This work was funded by the Helmholtz Association's Initiative and Networking Fund (grant agreement no. VH-NG-1531).





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
