# Peer review of "Low and Consistent Asymmetry Parameters in Arctic and Mid-latitude Cirrus"

_EGUsphere, 2025_

## Author Comment (AC1)

**Response to Reviewer #1**

We thank the reviewer for their careful and insightful reading of our manuscript and for high-lighting important areas for clarification and improvement. We provide detailed responses below, with the reviewer's comments shown in **dark blue**.

**Major Comments**

**Reviewer:** This is an interesting study of considerable merit for its measurements of an important single scattering parameter of cirrus in a location where few measurements have previously been made. The asymmetry parameter is of secondary importance to the optical depth for calculating cloud reflectivity. It is nonetheless important to get right, as existing uncertainties can lead to errors of up to a factor of two.

I have some rather deep concerns about the analyses however, generally with regards to the aspect that the measurement techniques seem insufficiently justified, and also the reported results appear to be implausible.

We thank the reviewer for their positive assessment of the scientific relevance of our work. In response to the reviewer's concerns, we have revised the manuscript to provide a clearer and more detailed discussion of the asymmetry parameter retrieval method, including a more detailed discussion of the uncertainties in the retrieval. We believe these additions improve the transparency and scientific robustness of our methodology.

**Line 116**

**Reviewer:** A threshold of 0.1 ms is applied to remove "shattering events". Some justification seems necessary here. Is there a distinct mode in the interarrival times that would suggest there is a shattering mode? How can it be known that such events are not a result of natural turbulent clustering of particles, a well known phenomenon in clouds? If such events were included, would it affect the calculated values of g, optical depth, and all the other microphysical parameters? What I suggest here is plotting a spectrum of interarrival times, logarithmically binned, on a log-log plot (i.e. d n/dlog(tau)). If the spectrum has the property of scale invariance, namely the slope is nearly constant across interarrival times, including 0.1 ms, then the physics governing interarrival times at 0.1 ms should be anticipated to be the same at any other scale. If there is a scale break or distinct mode, then a better argument can be made that such filtering is justified.

The reviewer is correct that the 0.1 ms threshold was selected based on inspection of the inter-arrival time distributions. Figure 1 shows a log-log plot of interarrival times from all CIRRUS-HL research flights. While the main mode of the distribution typically lies between 1 and $1000\,\mathrm{ms}$, several flights exhibit a distinct secondary mode at much shorter timescales. This secondary mode appears consistently at interarrival times shorter than $0.1\,\mathrm{ms}$, corresponding to spatial separations of less than approximately $20\,\mathrm{mm}$ (assuming an aircraft speed of $200\,\mathrm{m\,s^{-1}}$).

Such short separations are unlikely to be maintained for a long periods of time, as electrostatic interactions tends to lead to rapid aggregation, forming a single aggregate particle rather than a temporally separated sequence of distinct ice crystals. Therefore, this distinct population at very short interarrival times is interpreted as resulting from shattering events, consistent with earlier studies. Field et al. (2006) reported a similar break in interarrival time distributions and used a comparable threshold to separate cirrus particles from shattering artefacts in optical array probe data. Our interarrival time filtering approach follows this established methodology.

We will include this interarrival time spectrum in the Supplementary Information along with annotated identification of the threshold. Importantly, please note that the interarrival threshold correction was only applied when constructing particle size distributions and calculating bulk properties (N, IWC), not for the asymmetry parameter g. Values of g were retrieved solely from manually classified intact particles and were unaffected by the interarrival filtering.

[Figure]

Figure 1: Log–log distribution of interarrival times ($\tau$) for detected particle triggers during CIRRUS-HL research flights (RF02-RF23). For some flights a distinct second mode is visible for interarrival times below 0.1 ms (highlighted with red vertical line), which is attributed to shattering events.

**Line 131**

**Reviewer:** Baker and Lawson (2006) focused on mid-latitude clouds, not Arctic clouds. The premise of this submission here is to consider latitudinal variations in microphysical and optical properties. What justification is there that the power-law behavior identified here for relating mass to area, obtained at mid-latitudes, can be applied to the Arctic?

We acknowledge the reviewer's concern regarding the use of an area-to-mass relationship derived from mid-latitude cirrus for our Arctic data. It is well established that different mass-dimensional relationships can lead to significant uncertainties in retrieved ice water content (IWC) and, consequently, in the effective radius. We deliberately chose to use an area-to-mass relationship rather than a conventional diameter-to-mass parameterisation because linking particle mass to projected area has been shown to reduce mass estimation errors by approximately 50% Baker and Lawson (2006). Since particle cross-sectional area is directly measured in our dataset — unlike particle maximum dimension, which often relies on assumptions about shape and orientation — this method offers a more accurate basis for mass retrieval.

To our knowledge, no studies currently provide area-to-mass relationships specific to Arctic

cirrus, making the Baker and Lawson parameterization the most suitable option available. In response to the reviewer's comment, we will explicitly note in the manuscript that this relationship is based on mid-latitude observations to ensure full transparency.

**Line 154**
**Reviewer:** The data is stated to be "manually cleaned" based on "intact" imaged particles. This sounds very unscientific. Can a more objective justification be described for what is being done to what?

[Figure]

Figure 2: Log–log distribution of interarrival times ($\tau$) for detected particle triggers during CIRRUS-HL research flight RF02.

We emphasize that manual screening of the dataset is necessary, as the inter-arrival time method alone cannot identify all shattering events. Figure 2 illustrates this limitation using the inter-arrival time histogram from research flight 2. The red dashed line marks the threshold of 0.1 ms, which separates the main particle population (longer inter-arrival times) from the shattering mode (shorter inter-arrival times). The shattering mode abruptly ends near 10 μs, corresponding to the electronics dead time of the PHIPS system—that is, the minimum temporal separation required to register two distinct trigger events. This dead time translates to a spatial separation of approximately 2 mm at typical jet aircraft speeds.

However, the PHIPS image frame covers an area of roughly 1.1 mm × 1.5 mm, which converted into inter-arrival times is shown as the shaded grey region in Fig. 2. Shattering fragments arriving within these distances—at spacings below the dead time limit—cannot be detected by

the inter-arrival algorithm alone. Visual inspection of the stereo images is therefore an important and practical complementary method to identify such events. As illustrated in Fig. 2 of the main text, manual classification using the stereo images is essential to reliably remove residual shattering artifacts.

However, we acknowledge the reviewer's concern and have clarified both the reasoning behind this shattering correction as well as the procedure. Manual inspection refers to the visual classification of individual trigger events using stereo-microscopic images from PHIPS to identify likely shattering events. Shattering is identified when multiple small fragments (typically smaller than 100 µm) appear simultaneously on one or both stereo image frames. Given that each image covers an area of approximately 1.5 mm × 1.1 mm, the presence of several unconnected particles within this small field of view is highly unlikely under natural cirrus conditions, where ice crystal number concentrations and clustering tendencies are generally low. In addition, possible shattering of larger crystals is identified when the particle morphology shows clearly broken or truncated shapes, e.g., when expected hexagonal facets are missing or irregular, or when boundaries appear jagged rather than smooth.

In order to convince the reviewer that shattering events are identifiable from the stereo-images, we have added below two examples of shattering events (Fig. 3), which together with more examples, will be added to the Supplementary information.

[Figure]

(a) Example with small fragments

(b) Shattering of a larger aggregate

Figure 3: Stereo-image examples from PHIPS illustrating different particle shattering events. Each row shows the stereo-image pair separated by 120°: (a) small fragments and (b) shattering of a larger aggregate. The inter-fragment spacing in these examples is significantly below the threshold limit of 2 mm that is calculated from the electronics dead time of 12 µs (the blue vertical line in Fig. 2). Each image covers a field of view of approximately 1.5 mm × 1.1 mm.

**Section 2.2.3**

**Reviewer:** This section needs much more detail. A point of particular concern regards the validity and uncertainty related to the assumption stated in Xu et al (2022) that about the assumption lying behind the "mean" statement on l. 174 that "This is achieved by exploiting the assumption that the forward diffraction and the refraction - reflection energies are asymptotically equal." First, it's worth considering the rant in Bohren and Clothiaux about how there is no refraction, reflection or diffraction - only scattering and interference. The distinctions between the three are entirely artificial. But more importantly from a measurement standpoint, per Jarvinen et al (2023), the polar nephelometer only measures scattering at angles between 18 degrees and 170 degrees, which for any conceivable cloud particle encompasses quite a lot less than half the total scattered energy justifying a straightforward mean of forward and side/back scattering. Perhaps this all makes sense. I'm not sure I understand the need for a Legendre series expansion as described in Xu et al (2022). But at the very least, a full justification, with error analysis, should be presented of this measurement that is core to the article.

Thank you for pointing this out. We are aware that the separation of scattering into diffraction, refraction, and reflection is not fundamental from the perspective of electromagnetic theory. As Bohren & Clothiaux (2006) emphasize, all scattering results from the interaction of electromagnetic waves with matter, and the distinction into separate components is introduced only within the geometric optics (GO) framework. However, in atmospheric particle optics, this decomposition is still widely used and practically useful. For particles with large size parameters $x = 2\pi r/\lambda > 100$, geometric optics becomes increasingly accurate in describing the angular distribution of scattered light and the partitioning of energy. At the same time, rigorous electromagnetic methods such as DDA or T-matrix become computationally unfeasible for such large and complex particles. We therefore rely on the GO-based separation in our analysis, as it allows us to reconstruct the full angular scattering function from the limited angular range covered by PHIPS, following the approach described by Xu et al. (2022).

We have expanded Section 2.2.3 significantly. The retrieval of $g$ from the polar nephelometer follows the method described by Xu et al. (2022), where full scattering phase functions are reconstructed from measurements at angles 18°–170° using a Legendre polynomial expansion. In this approach, the basic assumption is that at scattering angles greater than 18°, the measured signal is dominated by geometric optics (GO) contributions, namely refraction and reflection, with only minor contributions from diffraction. The unmeasured forward angular range below 18° is assumed to contain the majority of the diffraction peak. Based on this separation, the GO part of the angular phase function can be reconstructed over the full angular range using the measured data and the Legendre polynomial expansion.

The accuracy of the retrieval algorithm is primarily determined by the integration error, which largely depends on the characteristics of the phase function, such as if peaks or sharp changes in the intensity appear that cannot be well captured by the limited angular range. To investigate the smoothness of the measured phase function, Xu et al (2022) introduced the $C_p$ parameter. This parameter reflects the decay rate of Legendre expansion coefficients and can be directly calculated from the measured angular intensity. When $C_p > 0.4$, the phase function is sufficiently smooth for the Legendre expansion to converge rapidly, and the error in the retrieved asymmetry parameter remains below 0.001. This has been validated by numerical ray-tracing simulations for orientation averaged hexagonal columns and plates with different aspect ratios and with varying degrees of distortion. In our dataset, 96% measured averaged angular scattering function profiles had $C_p$ values above this threshold, indicating that the error introduced by the expansion is negligible compared to other sources of uncertainty, such as the instrumental uncertainty.

The instrumental uncertainty arises from variations in optical transmission and from differences

in the individual channel responses of the multi-anode photomultiplier tube (MAPMT). These differences were characterised by performing calibrations using glass beads with known phase function (Wagner et al., 2025) and the resulting maximum uncertainty in $g$ is 0.008. We have added this discussion to the manuscript and now provide a more complete justification and error analysis for the asymmetry parameter retrieval.

**Table 1:**
**Reviewer:** The microphysical measurements presented appear implausible for what they would imply for the reflectivity based on the thin cloud expression given by Eq. 8. Taking the reported median microphysical values of 3 mg/m3 for the IWC, 37 um for the effective radius, the mean optical depth for a cloud 1 km thick would be 0.12. If, much more generously the cloud were 3 km thick, then it would be 0.36. From Eq. 8, this quite thick cirrus, taking g = 0.727, would have a reflectivity of 0.05. From the ground, such physically thick clouds would be barely visible. Mostly one would see blue sky. This seems implausible given cirrus are certainly very visible in the Arctic in satellite measurements, with reflectivities I would guess 10 times as high.

We appreciate the reviewer's effort to place our in situ microphysical results in the context of bulk radiative visibility and remote sensing observations. To assess consistency, the optical depths implied by our measured ice water content and effective radius can be compared to independent Arctic remote sensing observations. Using representative cloud geometries (1.4–2 km thickness) reported for Arctic cirrus over Ny-Ålesund (Nakoudi et al., 2021), the same microphysical values yield optical depths of roughly 0.19 to 0.27. These are in line with ground-based lidar climatologies: Campbell et al. (2020) report that cirrus in the Alaskan subarctic generally have optical depths below 0.5, and Nakoudi et al. (2021) find median values between 0.08 and 0.2 at Ny-Ålesund. Thus, the optical depth range we infer is comparable to established Arctic measurements.

The derived reflectivity of order 0.05 for such optically thin cirrus (using Eq. 8 with g=0.727) is physically plausible. Cirrus with these properties can appear faint from the ground, yet still be detectable, especially under favorable contrast conditions (e.g., against a darker sky or when the sun is at a non-zenith angle). While we acknowledge the reviewer's skepticism regarding the visual detectability of these clouds, we note that the concern is based on qualitative reasoning without quantitative comparison to observed reflectivity. Our estimates, in contrast, are now supported by published optical depth observations.

We agree with the reviewer that comparing our in situ results with bulk optical properties is valuable. Importantly, PHIPS directly measures projected particle area from two separate views, which can be used to estimate extinction coefficients under the geometrical optics assumption (i.e., extinction efficiency of 2). To strengthen the connection to remote sensing metrics, we will include the median, mean, and percentile values of the extinction coefficient in Table 1, along with a discussion of corresponding optical depth estimates..

**Lines 412–421:**
**Reviewer:** This paragraph risks being a bit misleading as global climate models are dynamic. It could well be that a value of g that is too high means a low bias in reflectivity with substantial instantaneous radiative forcing impacts. But there is a feedback. With less reflected, more sunlight is transmitted, which by heating the ground could destabilize the atmosphere to create more clouds, offsetting the low reflection bias that is discussed here.

We thank the reviewer for pointing this out. We will revise this paragraph to clarify that besides tuning the GCMs, the potential bias could lead to dynamic feedbacks within the GCM that can potentially offset or amplify the original error. We will bring the example of more sunlight reaching the surface, which in turn can lead to increased convection and potentially to more low-level cloud

formation.

**References**

Field, P. R., A. J. Heymsfield, and A. Bansemer, 2006: Shattering and Particle Interarrival Times Measured by Optical Array Probes in Ice Clouds. *J. Atmos. Oceanic Technol.*, **23**, 1357–1371, `https://doi.org/10.1175/JTECH1922.1`.

Baker, B. A., & Lawson, R. P. (2006). In Situ Observations of the Microphysical Properties of Wave, Cirrus, and Anvil Clouds. Part I: Wave Clouds. *Journal of the Atmospheric Sciences*, **63**(12), 3160–3185. `https://doi.org/10.1175/JAS3807.1`

Nakoudi, K., Ritter, C., and Stachlewska, I. S. (2021): Properties of Cirrus Clouds over the European Arctic (Ny-Ålesund, Svalbard). *Remote Sensing*, **13**(22), 4555. `https://www.mdpi.com/2072-4292/13/22/4555`, `https://doi.org/10.3390/rs13224555`.

Campbell, J. R., Dolinar, E. K., Lolli, S., Fochesatto, G. J., Gu, Y., Lewis, J. R., et al. (2020): Cirrus Cloud Top-of-the-Atmosphere Net Daytime Forcing in the Alaskan Subarctic from Ground-Based MPLNET Monitoring. *Journal of Applied Meteorology and Climatology*, **60**(1), 51–63. `https://doi.org/10.1175/JAMC-D-20-0077.1`

---

## Author Comment (AC2)

**Response to Reviewer #2**

We thank the reviewer for the critical comments. We appreciate the emphasis on the importance of accurate asymmetry parameter (g) retrievals. However, we strongly disagree with the assertion that our methodology is "flawed" or that the scientific value of our results is significantly reduced. Below we address the points raised.

**1. Validity of GO–diffraction separation**

The reviewer expresses concern that our retrieval follows the methodology of Xu et al. (2022), which separates diffraction and geometric-optics (GO) contributions. We note that such a separation is not unique to our work but is inherent to all retrieval approaches based on in situ polar nephelometer data, since near-forward scattering cannot be directly measured. Earlier methods (e.g., Gerber et al., 2000; Auriol et al., 2001) assigned a fixed fraction of forward-scattered energy, derived from idealised hexagonal crystal calculations, to compensate for the missing range. Our approach represents a significant improvement: instead of prescribing the missing contribution a priori, we reconstruct the GO part of the phase function using a Legendre polynomial expansion of the measured scattering pattern, and only parameterise the diffraction peak. This makes our retrieval data-driven rather than model-imposed.

We acknowledge that the separation of diffraction and geometric optics (GO) is not exact from the perspective of electromagnetic theory. Nevertheless, for particles with large size parameters ($x > 100$), this decomposition is well established in atmospheric optics (e.g., Takano and Liou, 1989; Macke et al., 1996) and provides the only practical framework for reconciling theoretical scattering calculations with in situ measurements. Although improved GO methods have been developed to avoid this artificial separation, they have been shown to converge toward conventional ray-tracing results for the large size parameters relevant to our study (Yang and Liou, 1996), which further justifies the use of this approach. Importantly, the GO–diffraction separation is not unique to interpretation of in situ measurements but underlies many conventional ray-tracing schemes and remains the foundation of widely applied parameterisations and retrieval algorithms (e.g., Baran and Labonnote, 2007; van Diedenhoven et al., 2012). Rejecting this separation would therefore not only undermine our methodology, but also invalidate a large body of established research in atmospheric light scattering.

**2. On forward peak and delta-transmission**

The reviewer raises the concern that a narrow forward peak, caused by so-called delta rays in smooth crystals, could bias the retrieved values of $g$. We note that such features require pristine, plane-parallel crystal surfaces. Our measured phase functions, however, show no indications of smooth-surface scattering such as distinct 22° halos or strong specular reflections. Instead, the measured orientation-averaged angular scattering functions (Fig. 1) are consistent with complex or roughened surfaces. As demonstrated in Fig. 1, the delta-transmission peak is even more sensitive to surface roughness than halo or specular features, and it disappears at comparatively low levels of distortion. Consequently, it is highly improbable that delta-transmission features were present in the observed angular scattering functions, and thus unlikely that our retrievals of $g$ are biased by this effect.

[Figure]

Figure 1: Statistical analysis of all PHIPS-measured orientation averaged phase functions that were used to retrieve g. The median function (shown as black circles) shows smooth angular behavior and no sharp delta features are present in the inter-quartile or in the 1 to 99 percentile range (shown as shaded areas), consistent with roughened ice particles. Overlaid are ray tracing simulations of the geometrical optics part of the phase function (without diffraction) for a hexagonal crystal with a unity aspect ratio and the length of 100 µm. The solid lines show the full phase function and circular markers show the phase function integrated over the PHIPS detection geometry. It can be seen that the delta peak vanishes at roughness of $\sigma$=0.1, whereas measurable halo-features are still seen. The unity model is not the best fit to the measurement data but demonstrates that $\sigma$ values above 0.1 are needed to reproduce the measured featureless function.

**3. Use of truncated Legendre expansion**

The reviewer is concerned that truncation of the Legendre expansion may bias the results by excluding narrow forward peaks associated with smooth, pristine crystals, which would in turn lead to underestimation of $g$. We stress that this is not the case for our analysis. Truncation is not an arbitrary choice but a necessary step to avoid overfitting the limited angular range covered by PHIPS. More importantly, Xu et al. (2022) showed with ray-tracing simulations (their Figures 5 and 6) that when the expansion is truncated at orders consistent with the measurement range, the reconstruction error in $g$ remains typically below 0.001 for roughened hexagonal crystals.

Crucially, our measurements do not contain smooth planar crystals. Instead, the measured phase functions are consistent with rough or complex particle surfaces. This is further supported by the $C_p$ values in our dataset, which exceed 0.4 in nearly all cases (Fig. 2), indicating that the method is being applied well within its validity range.

Thus, while we fully acknowledge that the Legendre expansion is not applicable to smooth particles with narrow forward peaks, we explicitly do not need to worry about this regime in our analysis. Within the relevant parameter space of our measurements, the associated error in $g$ is small and quantitatively constrained.

**4. Diffraction approximation**

The reviewer is concerned that our use of scalar diffraction for circular apertures underestimates the effect of noncircular particle shapes. However, orientation-averaged diffraction is primarily a

[Figure]

Figure 2: Distribution of $C_p$ values from the PHIPS dataset. The vertical dashed line marks the decay rate of the expansion coefficients $C_p = 0.4$. Nearly all observations lie above this threshold, implying $g$ retrieval errors below 0.001.

function of the projected area (Babinet's principle), and differences between circular and hexagonal apertures average out for large, randomly oriented ice crystals. Numerous studies (Liu and Yao, 1996; Hesse, 2008, and references therein) have shown that the diffraction peak is confined to angles well below the PHIPS cutoff of 18°, such that its detailed shape does not influence our retrieval of g. We therefore maintain that the circular-aperture approximation introduces only negligible error into the asymmetry parameter.

**5. On the request for a "health warning"**

We respectfully disagree with the reviewer that our results require a warning label suggesting unreliability. Our methodology represents a well-founded improvement over previous approaches, is supported by numerical validation (Xu et al., 2022), and includes explicit uncertainty estimates. All approaches to ice cloud optics—whether based on theory, laboratory experiments, or in situ data—are necessarily approximate, given the complexity of real atmospheric ice. Our contribution lies in advancing the experimental basis for g, providing observational constraints that are otherwise sparse. We believe this substantially increases, rather than decreases, the scientific value of the manuscript.

**6. Conclusion**

In summary, the reviewer's criticisms appear to dismiss not only our approach but, by implication, all experimental efforts to derive g for large, irregular ice particles. We maintain that our method represents a significant advance: it combines measured scattering with a physically justified reconstruction framework, avoids the ad hoc assumptions of earlier retrievals, and provides explicit, quantitative error bounds. Our results therefore contribute an essential and meaningful step forward in establishing an experimental basis for cirrus radiative properties. We do not argue that our approach is the only way to derive g from our observations and welcome future modeling efforts to

test and complement our in-situ data with methods beyond our current computational reach.

**References**

Xu, G., M. Schnaiter, and E. Järvinen, 2022: Accurate Retrieval of Asymmetry Parameter for Large and Complex Ice Crystals From In-Situ Polar Nephelometer Measurements. *J. Geophys. Res. Atmos.*, **127**, e2022JD036608, `https://doi.org/10.1029/2022JD036608`.

Auriol, F., J.-F. Gayet, O. Crepel, A. Fournol, and S. Oshchepkov, 2001: In situ observation of cirrus scattering phase functions with 22° and 46° halos: Cloud field study on 19 February 1998. *J. Geophys. Res.*, **106**, 17923–17931, `https://doi.org/10.1029/2000JD900762`.

Gerber, H., Y. Takano, T. J. Garrett, and P. V. Hobbs, 2000: Nephelometer measurements of the asymmetry parameter, volume extinction coefficient, and backscatter ratio in Arctic clouds. *J. Atmos. Sci.*, **57**, 3021–3034, `https://doi.org/10.1175/1520-0469(2000)057<3021:NMOTAP>2.0.CO;2`.

Macke, A., J. Mueller, and E. Raschke, 1996: Single scattering properties of atmospheric ice crystals. *J. Atmos. Sci.*, **53**, 2813–2825, `https://doi.org/10.1175/1520-0469(1996)053<2813:SSPOAI>2.0.CO;2`.

Takano, Y., and K.-N. Liou, 1989: Solar radiative transfer in cirrus clouds. Part I: Single-scattering and optical properties of hexagonal ice crystals. *J. Atmos. Sci.*, **46**, 3–19, `https://doi.org/10.1175/1520-0469(1989)046<0003:SRTICC>2.0.CO;2`.

Takano, Y., and K.-N. Liou, 1996: Geometric-optics–integral-equation method for light scattering by nonspherical ice crystals. *Appl. Opt.*, **35**, 6568–6584, `https://opg.optica.org/abstract.cfm?URI=ao-35-33-6568`.

Baran, A. J., and L.-C. Labonnote, 2007: A self-consistent scattering model for cirrus. I: The solar region. *Quart. J. Roy. Meteor. Soc.*, **133**, 1899–1912, `https://doi.org/10.1002/qj.164`.

van Diedenhoven, B., B. Cairns, I. V. Geogdzhayev, A. M. Fridlind, A. S. Ackerman, P. Yang, and B. A. Baum, 2012: Remote sensing of ice crystal asymmetry parameter using multi-directional polarization measurements—Part 1: Methodology and evaluation with simulated measurements. *Atmos. Meas. Tech.*, **5**, 2361–2374, `https://doi.org/10.5194/amt-5-2361-2012`.

Liu, C., and K. Yao, 1996: Calculation of ice crystal diffraction. *Adv. Atmos. Sci.*, **13**, 340–348, `https://doi.org/10.1007/BF02656851`.

Hesse, E., 2008: Modelling diffraction during ray-tracing using the concept of energy flow lines. *J. Quant. Spectrosc. Radiat. Transfer*, **109**, 1374–1383, `https://doi.org/10.1016/j.jqsrt.2007.11.002`.

---

## Author Comment (AC3)

**Response to Reviewer #3**

We thank the reviewer for raising this important point. A closure study of the type requested has already been presented in Xu et al. (2022, their Fig. 6). That analysis showed that for roughened hexagonal crystals the retrieval error in $g$ is typically below 0.001 when the complexity parameter $C_p > 0.4$. We acknowledge, however, that the range of crystal configurations considered in the original study was limited.

Since then, we have substantially extended our ray-tracing database using the code of Macke et al. (1996). The new simulations cover hexagonal crystals within the size range from 20 to 500 µm and for aspect ratios from 0.68 to 0.85 that reflect the microphysical results from the PHIPS stereo-imaging. Surface complexity was introduced by applying tilted-facet Gaussian roughness with $\sigma$ between 0 and 0.9, and in some cases additional complexity was included by implementing the mean-free-path model to simulate internal air inclusions. In total, 1612 different orientation-averaged crystal configurations were generated.

For each case, the simulated phase function was truncated to the PHIPS angular range, integrated over the PHIPS detector geometry, and passed through the Xu et al. retrieval algorithm. The retrieved $g$ was then compared against the true value from the Macke code. Results for single-habit simulations (Fig. 1) show that while many cases reproduce $g$ within the $\pm 0.001$ uncertainty reported in Xu et al. (2022), some configurations produce larger deviations, occasionally exceeding 0.05 even for $C_p > 0.4$. The bias is generally positive, confirming earlier reviewer concerns that the method may underestimate $g$. However, the fraction of simulations that have a bias of larger than 0.02 is around 18%. Further, we stress that such single-habit cloud particle populations are not representative of natural cirrus populations, which consist of mixtures of habits, aspect ratios, and roughness states.

[Figure]

(a) Difference between true and retrieved $g$ vs. retrieved $C_p$. Red dashed lines indicate $\pm 0.001$.

[Figure]

(b) Histogram of the retrieval error in $g$ for 1612 simulations of single habits.

Figure 1: Closure test of $g$ retrieval using 1612 simulated phase functions of individual crystal habits.

To reflect more realistic cloud conditions, we repeated the closure test using ensembles of 10 randomly selected crystal morphologies. As shown in Fig. 2, the ensemble results converge rapidly, with retrieval errors consistently below 0.02. Increasing the number of morphologies confine the results even further below 0.02. The bias remains predominantly positive, indicating that the method tends to underestimate $g$, again consistent with the reviewer's expectation.

[Figure]

[Figure]

(a) Difference between true and retrieved $g$ vs. retrieved $C_p$.

(b) Histogram of retrieval error in $g$ for ensembles of 10 habits.

Figure 2: Closure test of $g$ retrieval using ensembles of 10 randomly selected crystal morphologies.

Based on these expanded closure studies, we adopt a conservative updated uncertainty estimate of about $+0.02$ in $g$. We will include this revised estimate, together with a clearer and more comprehensive description of the retrieval methodology, in the manuscript, as already indicated in the answers to the other reviewers. Importantly, even with this uncertainty the central result of our study remains unchanged: $g$ in both Arctic and mid-latitude cirrus is systematically lower than the values typically assumed in current ice particle optical models (0.76-0.88) (e.g. Iaquinta et al., 1995; Yang et al., 2008; Um and McFarquhar, 2009).

**References**

Iaquinta, J., H. Isaka, and P. Personne, 1995: Scattering Phase Function of Bullet Rosette Ice Crystals. *J. Atmos. Sci.*, **52**, 1401–1413, `https://doi.org/10.1175/1520-0469(1995)052<1401:SPFOBR>2.0.CO;2`.

Macke, A., J. Mueller, and E. Raschke, 1996: Single scattering properties of atmospheric ice crystals. *J. Atmos. Sci.*, **53**, 2813–2825, `https://doi.org/10.1175/1520-0469(1996)053<2813:SSPOAI>2.0.CO;2`.

Um, J., and G. M. McFarquhar, 2009: Single-scattering Properties of Aggregates of Plates. *Q. J. R. Meteorol. Soc.*, **135**, 291–304, `https://doi.org/10.1002/qj.378`.

Yang, P., Z. Zhang, G. W. Kattawar, S. G. Warren, B. A. Baum, H.-L. Huang, Y. X. Hu, D. Winker, and J. Iaquinta, 2008: Effect of Cavities on the Optical Properties of Bullet Rosettes: Implications for Active and Passive Remote Sensing of Ice Cloud Properties. *J. Appl. Meteor. Climatol.*, **47**, 2311–2330, `https://doi.org/10.1175/2008JAMC1905.1`.

Xu, G., M. Schnaiter, and E. Järvinen, 2022: Accurate Retrieval of Asymmetry Parameter for Large and Complex Ice Crystals From In-Situ Polar Nephelometer Measurements. *J. Geophys. Res. Atmos.*, **127**, e2022JD036608, `https://doi.org/10.1029/2022JD036608`.